# Online Signature Analysis for Characterizing Early Stage Alzheimer’s Disease: A Feasibility Study

**DOI:** 10.3390/e21100956

**Published:** 2019-09-29

**Authors:** Zelong Wang, Majd Abazid, Nesma Houmani, Sonia Garcia-Salicetti, Anne-Sophie Rigaud

**Affiliations:** 1Electronics and Physics Department, Telecom SudParis, Institut Polytechnique de Paris, 9 rue Charles Fourier, 91011 Evry, France; 2SAMOVAR, Telecom SudParis, CNRS, Institut Polytechnique de Paris, 9 rue Charles Fourier, 91011 Evry, France; 3AP-HP, Groupe Hospitalier Cochin Paris Centre, Hôpital Broca, Pôle Gérontologie, 75013 Paris, France; 4L’unité de Recherche Universitaire EA 4468, Université Paris Descartes, 75006 Paris, France

**Keywords:** online signature analysis, sample entropy, Alzheimer’s disease, classification

## Abstract

We aimed to explore the online signature modality for characterizing early-stage Alzheimer’s disease (AD). A few studies have explored this modality, whereas many on online handwriting have been published. We focused on the analysis of raw temporal functions acquired by the digitizer on signatures produced during a simulated check-filling task. Sample entropy was exploited to measure the information content in raw time sequences. We show that signatures of early-stage AD patients have lower information content than those of healthy persons, especially in the time sequences of pen pressure and pen altitude angle with respect to the tablet. The combination of entropy values on two signatures for each person was classified with two linear classifiers often used in the literature: support vector machine and linear discriminant analysis. The improvements in sensitivity and specificity were significant with respect to the a priori group probabilities in our population of AD patients and healthy subjects. We show that altitude angle, when combined with pen pressure, conveys crucial information on the wrist-hand-finger system during signature production for pathology detection.

## 1. Introduction

Signatures have long been used for personal biometric authentication. This behavioral modality, widely used in our daily lives, evolves with age. The production of fine motor movements strongly progresses during childhood and for many years until the motor program ruling handwritten movements reaches maturity. At this stage, the signing gesture becomes the outcome of a highly automated ballistic process, involving multiple areas of the brain [1]. Alterations in this automated and unconscious process may reflect a change in health condition, such as a decline in cognitive or motor functions.

Few researchers have examined handwriting signatures for assessing the health state of individuals. Prior studies on handwriting focused on other handwriting tasks, such as loops, letters, and handwritten texts, for characterizing and detecting different pathologies, such as Parkinson’s disease [2,3,4,5,6,7,8,9], Huntington’s [9], Alzheimer’s [9,10,11,12,13,14,15,16], and depression [10].

Alzheimer’s disease (AD), from the perspective of handwritten signatures analysis, has rarely been studied. To the best of our knowledge, only one study, specifically on online handwritten signatures [17], proposed a machine learning approach to discriminate AD patients from healthy subjects. The authors exploited the sigma-lognormal model [18] for online signatures to extract global features of each signature instance, such as the number of lognormal densities, the maximum speed divided by the signing time, and the number of peaks in the graphing speed normalized by their duration, average, and standard deviations of some key parameters of the sigma-lognormal model. Three classifiers were compared: a support vector machine (SVM) with a linear kernel, and two decision trees: classification and regression trees (CART) and bagging CART. The best results were obtained with the bagging CART algorithm; an error rate of 3% was reached on a private database of 32 healthy and 29 pathological signatures.

In the present work, we studied the relationship between handwriting movements and early-stage Alzheimer’s disease (AD) by analyzing online signatures captured in a real-life task: signing a check. We propose considering the raw temporal functions acquired by the digitizer and using the sample entropy (SE) measure to quantify the possible alterations in temporal functions induced by the disease in the early stages. Our hypothesis is that since signing is uniquely both an automated and an unconscious writing pattern, a cognitive and motor disorder during the early stage of disease would be detectable by alterations in the raw temporal functions. This analysis may provide insights for early AD diagnosis.

Our paper is organized as follows: In Section 2, we detail the database and its associated acquisition protocol. In Section 3, we review the SE measure and present the experiments and results. Section 4 states our conclusions and perspectives for future work.

## 2. Materials and Data Acquisition Protocol

Our private dataset was acquired at Broca Hospital in Paris within the framework of the ALWRITE project [15,16], a French research project on the analysis of different handwriting tasks for AD assessment. All participants freely signed a consent form after receiving information about the study. The whole protocol contained, among many others, a real-life-inspired task that involves filling out a simulated check printed on a sheet of paper and fixed to a digitizing tablet.

Two simulated checks (pseudo-checks) were provided on two different sheets of paper and presented consecutively to each participant. Each person naturally filled out the whole information in the check by appending the amount in words and numeric form, the date, the place, and their signature. Participants filled out both pseudo-checks on a Wacom Intuos Pro Large tablet with an inking pen. The digitizing tablet sampled the pen trajectory at 125 Hz and captured five raw temporal functions: pen position (*x*(*t*),*y*(*t*)), pen pressure *p*(*t*), and two pen inclination angles, azimuth *Az*(*t*) and altitude *Alt*(*t*), as shown in Figure 1.

For our study, online signature data were manually segmented from the whole handwritten information written on the check (the amount in words, the amount in numeric form, the date, and the place). Examples of online signatures from one healthy subject and one early-stage AD patient are displayed in Figure 2. As the in-air trajectory of the pen (pen-up) is also captured by the digitizer up to 2 cm off the tablet surface, the signatures displayed in Figure 2 also show in-air trajectories in green points.

Figure 3, Figure 4, Figure 5, Figure 6 and Figure 7 display the five raw temporal functions for one AD signature, and a healthy one, belonging to the persons whose signatures are shown in Figure 2. We notice a different behavior between these two signatures in terms of their raw temporal functions.

The used dataset contained online signatures of 70 participants. From the overall dataset, 31 participants were patients with early-stage AD (AD patients), and 39 were healthy controls (HC subjects). Table 1 summarizes the characteristics of the subjects. AD patients were diagnosed at Broca Hospital (Paris, France) based on the DSM-V criteria [19] and considered as having early-stage AD if their mini-mental state examination (MMSE) score was over 20 [19]. HC subjects performed neuropsychological tests to ensure their cognitive profile was normal. Participants with medical problems, such as stroke and other neurodegenerative diseases, were not included.

## 3. Methods and Analysis of Signatures

### 3.1. Sample Entropy Measure

The concept of entropy has been widely and successfully exploited in the literature for the analysis of biomedical signals, especially for electroencephalography analysis [20,21]. More recently, SE was used on online handwritten inputs for health status assessment [22,23]. It is used to quantify irregularities in time sequences and measures complexity. To the best of our knowledge, SE has never been used for AD assessment based on signature analysis.

On-time series of length *N* {*u*(1), *u*(2),…, *u*(*i*),…, *u*(*N*)}, an integer value *m* < *N* is fixed, specifying the length of the subsequences to be compared (window on the signal), and let *r* be a positive real number.

The following (*N* − *m* + 1) vectors *X_m_*(1), *X_m_*(2), …, *X_m_*(*i*), …, *X_m_*(*N* − *m* + 1) are defined as:
(1)$$X_{m}\left(i\right)=\{u\left(i\right),u\left(i+1\right),\ldots ,u\left(i+m-1\right)$$
and are embedded in the *m*-dimensional space, since each contains a subsequence of length *m* starting at point *i*. The distance between two different vectors (*i* ≠ *j*) is defined as the maximum difference of their scalar components:
(2)$$d\left(X_{m}\left(i\right),X_{m}\left(j\right)\right)=max_{0\leq k\leq m-1}\left\{\left|u\left(i+k\right)-u\left(j+k\right)\right|\right\}$$


SE is then:
(3)SE(m,r)=−log (A/B)
where *A* is the number of template vectors pairs less than *r* far in the embedded dimension (*m* + 1), and *B* is the number of template vectors pairs less than *r* far in the embedded dimension *m*. With construction *A* ≤ *B*, we have SE(m,r)≥0.

In the literature [20,21,22,23], the hyper-parameters *m* and *r* are often fixed empirically with the objective to maximize the discrimination between the groups: between Parkinson patients and healthy subjects [22], between AD patients and healthy subjects [20,21], and between patients with essential tremor and healthy subjects [23]. In this work, we studied the variation in SE values for all possible values in a given interval for both hyper-parameters *m* and *r*. The objective was to analyze the behavior of this measure independently of fixed values of its hyper-parameters.

### 3.2. Initial Trends in the Correlation Between SE and Metadata

We computed the SE values of all signatures for different values of *m* = 1, …, 9 and *r* = 0.1 to 0.9 in 0.1 steps. The SE of a signature was computed considering the whole signing gesture, including both in-air and on-paper trajectories during the signing process.

To gain initial insights into its behavior, we computed the SE for each raw temporal function separately. Then, we analyzed the correlation between the obtained values and two metadata, the MMSE and age, and that for the 70 participants independently of the group (AD or HC).

The correlation values between the MMSE and the average SE on both signatures of each person for all values of *m* and *r* are reported in Table 2. We noticed a positive correlation between the MMSE and SE that was higher for the *y* coordinate, pen pressure, and altitude angle compared with those computed for *x* coordinate and azimuth angle. This result shows that persons with normal cognitive profiles (high MMSE) tend to produce signatures with more information content.

Table 3 displays the correlation values between age and the average SE on both signatures for all values of *m* and *r*. We noticed a negative correlation between age and SE for each temporal function. This reveals that, in general, with age, entropy tends to decrease, meaning that temporal functions show less intrinsic irregularities and variation in the signature gesture. The handwritten production loses information content with age.

We observed a very low correlation between age and azimuth compared with other temporal functions (−0.1). Aging appears to have no effect on this parameter. Azimuth could be used to distinguish a right-handed from a left-handed person, for example, for biometric authentication purposes. In our dataset, except one person, all the other participants were right-handed and thus had a similar azimuth profile. Therefore, our dataset had less variance in azimuth between persons compared with all other temporal functions. This is why we did not consider the azimuth temporal function in the following studies.

### 3.3. Influence of Signature Styles

SE quantifies the intrinsic information content in signatures, and thus, may be sensitive to signature style. For this reason, we addressed whether HC and AD patients produced different signature styles and studied the influence of style on SE.

Signatures of HC and AD patients were labeled visually by three experts into text-based signatures (where all the allographs are legible), mixed (where one or more but not all of the allographs are legible), and stylized (where none of the allographs are legible) [24]. These three styles may be related to signature complexity, quantified by SE. The resulting distribution in each style is reported in Table 4.

We first noticed that text-based signatures do not exist in our dataset; no signatures were completely legible.

To study the influence of signature stylization, we considered the SE values computed for the *x* coordinate (SE(*x*)) and *y* coordinate (SE(*y*)), and that for all signatures, independent of the group (AD or HC). Then, we performed *k*-means clustering, with *k* = 3, on both entropy values simultaneously (two-dimensional (2D) clustering). Table 5 reports the distribution of signature styles in the obtained three groups and their respective bi-dimensional centroids.

We observed that 77.14% of persons with stylized and mixed signatures were grouped together in Group 2 (54 of the 70 participants). This means that most signatures with such styles have similar information content in the *x*(*t*) and *y*(*t*) temporal functions. Also, two persons with stylized signatures had the highest SE values, which means that an illegible signature could be complex.

These results indicate that SE quantifies irregularities in temporal functions independent of the stylization in the form. Given these findings, we then studied the relationship between SE and AD independent of signature styles.

### 3.4. Relationship between SE and AD

In each group (AD vs. HC), we analyzed the impact of the hyper-parameters *m* and *r* on SE values computed for the four temporal functions *(x*(*t*), *y*(*t*), *p*(*t*), and *Alt*(*t*)). In the surfaces displayed in both Figure 8 and Figure 9, we noticed that the SE measure was sensitive to the hyper-parameter *r* and not to *m*. Also, for low values of *r*, SE values were much higher for HC compared with AD patients. This means that the intrinsic variation in the temporal functions in the signature was much stronger for healthy persons compared with AD patients. We, thus observed a significant information loss in Alzheimer’s signatures for low values of hyper-parameter *r* in Figure 8 (for *x*(*t*) and *y*(*t*) pen coordinates) and Figure 9 (for pen pressure *p*(*t*) and altitude angle *Alt*(*t*)).

To gain further insight into the impact of the two hyper-parameters on SE, Figure 10, Figure 11, Figure 12 and Figure 13 display boxplots of the entropy values for the four temporal functions when fixing one hyper-parameter and varying the other. These figures show that SE is more sensitive to *r* compared with *m*. We observed that when *r* increases, entropy values decrease for both AD and HC groups and their variances for persons in each group decrease. When *m* increases, the entropy values and their variances for persons in each group tend to remain stable.

We also noticed that entropy values are much higher for temporal function *y*(*t*) compared with *x*(*t*), indicating that signatures vary more vertically than horizontally. The same occurs for altitude: it had higher information content than horizontal movement (*x*(*t*)). The altitude angle conveys information about how the writer holds and moves the pen when signing, and is particularly modified by the intensity of vertical movement (*y*(*t*)). Therefore, strong variations in *y*(*t*) also appear in the temporal function *Alt*(*t*).

Concerning information loss in Alzheimer’s signatures relative to healthy subjects, observed for low values of *r* in Figure 8 and Figure 9, the phenomenon was clearly stronger for pen pressure and pen altitude angle, particularly for *m* = 3 and *r*/*std* = 0.1. For pen pressure and pen altitude, we found a relative increase of entropy of around 50% and 37% between signatures of AD patients and those of healthy subjects, respectively.

### 3.5. Towards AD Detection: AD vs. HC Classification Results

To study the discriminative power of SE between AD and HC subjects, as SE is sensitive to *r*, we performed classification for *m* = 3 and *r*/*std* = 0.1. For these values of hyper-parameters, the Mann–Whitney test in Table 6 shows a significant difference between the entropy values of AD and HC populations for the four temporal functions.

Based on this result, we considered two linear classifiers, a linear support vector machine (SVM) and linear discriminant analysis (LDA), using a two-fold cross-validation protocol for training and testing. For each raw temporal function considered, the SE values of both signatures for each person were used as inputs to the classifier.

Table 7 and Table 8 report the obtained sensitivity (percentage of AD patients correctly classified), specificity (percentage of HC correctly classified), and accuracy values for the SVM and LDA classifiers.

We noticed that for both SVM and LDA classifiers, the combination of SE values computed for pen pressure and pen altitude improved the sensitivity relative to taking each of those features separately. As the a priori group probability for the dataset here used was 0.44 for the AD group (31 AD patients out of 70) and 0.56 for the HC group (39 subjects out of 70), the improvement obtained was significant. For sensitivity, which was our main objective, the relative improvement was 45.7% with both classifiers (64.52% in sensitivity relative to 44.28% of a priori Alzheimer’s group probability).

When combining the SE values of the four temporal functions, the sensitivity (up to 74.19%) and specificity (up to 76.92%) improved significantly for the SVM classifier. For the LDA classifier, sensitivity also improved (up to 67.74%) but the specificity decreased by almost 26%.

Thus, in the best-case scenario, with the SVM and the combination of the four SE values, the relative improvements in sensitivity and specificity were 67.54% and 37.35% relative to the a priori group probability of correct classification of AD patients and HC subjects, respectively.

## 4. Discussion and Conclusions

In this work, we analyzed the sample entropy (SE) of raw temporal signals captured by a digitizer on signatures appended during a simulated check-filling task. Our extensive study of the effect of the hyper-parameters when measuring SE added to a statistical analysis highlighted that online signatures written by early-stage AD patients show significantly lower entropy values, and thus, lower information content, than those of healthy persons. We verified that SE is not influenced by different signature styles, only represented in our dataset by mixed and stylized signatures [24].

We observed that the difference in the information content between signatures of AD patients and those of HC was high for pen pressure and altitude angle time sequences. This result that the pen altitude angle allows characterizing the pathology with pen pressure on signatures is a novel finding. Notably, in former work on biometrics, pen inclination angles were found to be misleading for characterizing a writer (with or without pressure) [25]. Our analysis instead showed that AD patients have less tonus in their way of holding the pen, which impacts pen pressure and variations in the vertical direction, which are lower, compared with those of healthy subjects. Our classification results suggest that altitude, when combined with pen pressure, conveys valuable information about the wrist-hand-finger system during signature production for pathology detection. In other words, one of our findings is that the way the pen is held by the writer during the signing gesture has significance for AD detection. Our statistical analysis with the Mann–Whitney test showed that a significant difference exists between AD and HC distributions of SE values for pen coordinates, pressure, and altitude angle. When combining these four temporal functions, we obtained an accuracy of 75.71% with a linear SVM classifier in the framework of a two-fold cross-validation protocol to ensure the reliability of the classification results.

Finally, the experiments and analyses conducted to support our claim—quantifying the alterations in raw temporal functions with SE, describing the complete signing gesture in the air and on paper, provides insight into early AD diagnosis. Our study shows that online signature, despite being a signal of short duration, conveys valuable information about a writer’s motor and cognitive skills through its ballistic, automated, and unconscious process, and is thus a possible modality for the automatic health state assessment.

However, our study has some limitations: (1) our dataset is small, both in terms of the number of persons in each group (AD and HC), and the number of signatures per person; (2) we exploited only the raw temporal functions captured by the digitizer for the data description; and (3) even if we showed that a statistically significant difference exists between AD and HC populations with the SE measure, we obtained a misclassification rate of around 24% with a linear SVM classifier. The rather low correlation coefficients between SE and the MMSE suggest that SE is not sufficient alone to finely characterize the alterations in temporal functions induced by the disease.

In the future, we will address these limitations by (1) incorporating other online signatures captured during another task in the ALWRITE project; (2) considering other kinematic parameters such as speed, acceleration, and jerk, successfully used for AD assessment based on handwriting after a local point-wise feature extraction [15,16]; and (3) combining SE with other non-linear measures to propose a more refined detection of alterations in signatures due to the disease.

We also aim to extend this transversal study to signatures acquired in different sessions. The longitudinal framework will contribute to evaluating the viability of our approach for quantifying the degradations in signatures according to dementia severity.

## Figures and Tables

**Figure 1 entropy-21-00956-f001:**
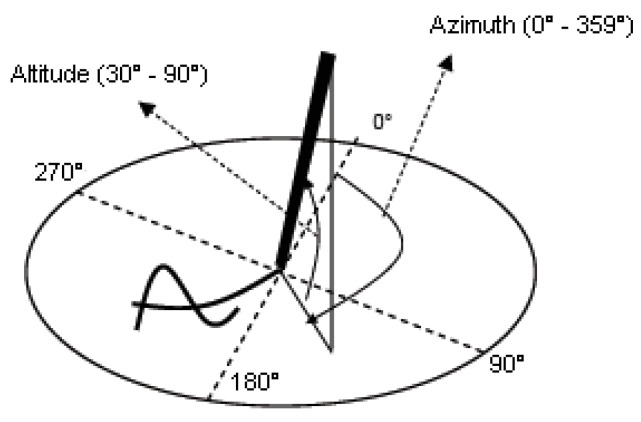
Azimuth and altitude angles captured by the Wacom digitizing tablet.

**Figure 2 entropy-21-00956-f002:**
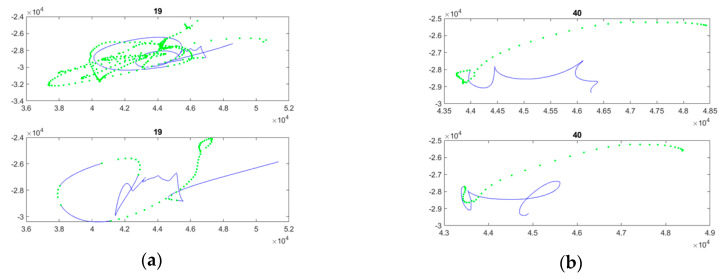
Examples of segmented signatures: (**a**) both signatures of an Alzheimer’s disease (AD) patient; (**b**) both signatures of a healthy subject. Green points represent the in-air trajectory captured by the Wacom tablet.

**Figure 3 entropy-21-00956-f003:**
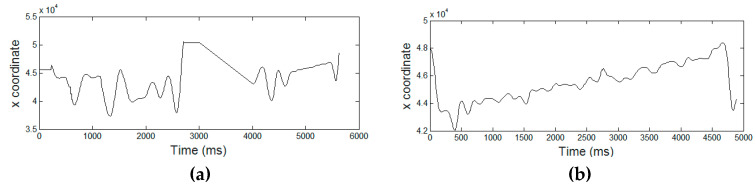
The *x* coordinate temporal function for (**a**) an AD signature and (**b**) a healthy one.

**Figure 4 entropy-21-00956-f004:**
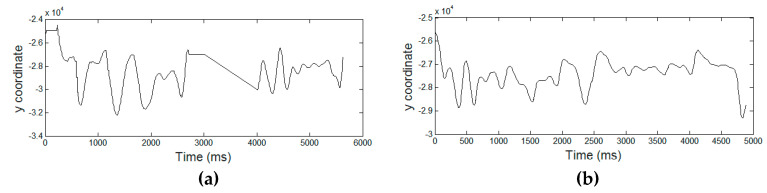
The *y* coordinate temporal function for (**a**) an AD signature and (**b**) a healthy one.

**Figure 5 entropy-21-00956-f005:**
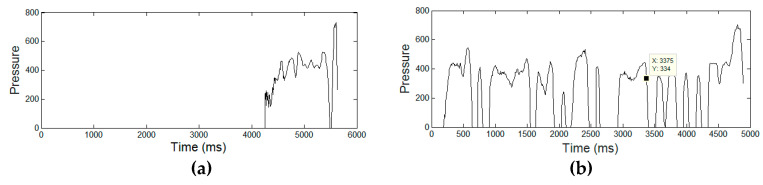
The pressure temporal function for (**a**) an AD signature and (**b**) a healthy one.

**Figure 6 entropy-21-00956-f006:**
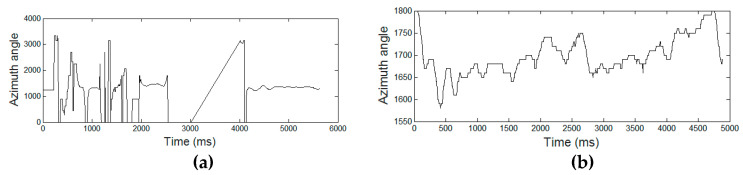
The azimuth angle for (**a**) an AD signature and (**b**) a healthy one.

**Figure 7 entropy-21-00956-f007:**
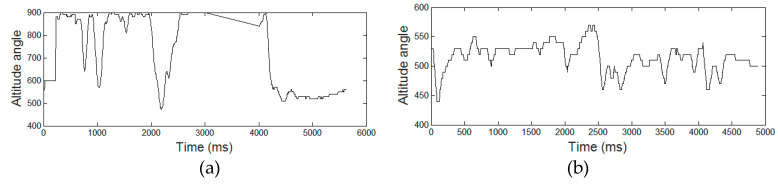
The altitude angle for (**a**) an AD signature and (**b**) a healthy one.

**Figure 8 entropy-21-00956-f008:**
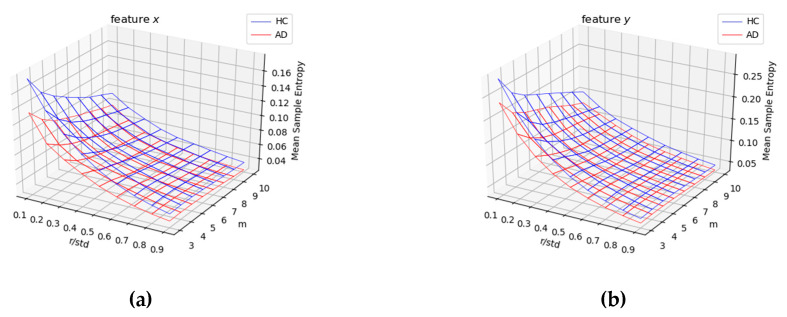
Average SE values on AD patients (red) vs. HC subjects (blue) for different values of the hyper-parameters *m* and *r* (*r* divided by the standard deviation (*std*) of the corresponding raw signal) considering the (**a**) *x*(*t*) and (**b**) *y*(*t*) coordinates.

**Figure 9 entropy-21-00956-f009:**
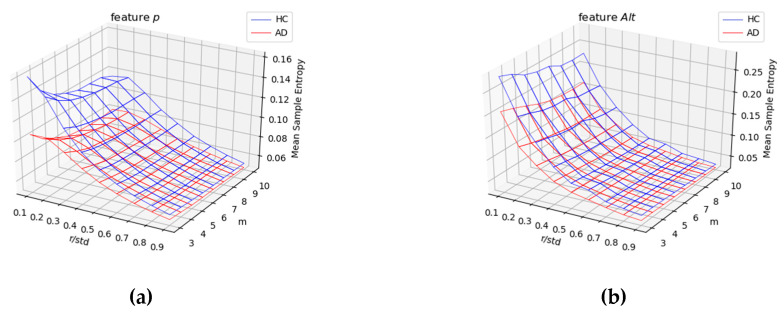
Average SE values on AD patients (red) vs. HC subjects (blue) for different values of the hyper-parameters *m* and *r* (*r* divided by the standard deviation (*std*) of the corresponding raw signal) considering: (**a**) pen pressure *p*(*t*) and (**b**) pen altitude angle *Alt*(*t*).

**Figure 10 entropy-21-00956-f010:**
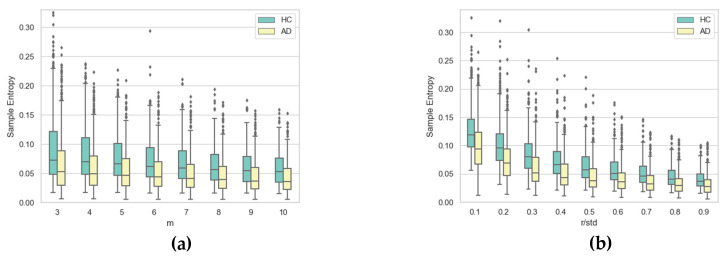
Boxplots of SE values for AD and HC considering *x*(*t*) coordinate for (**a**) each value of *m* when varying *r* and (**b**) each value of *r* when varying *m*.

**Figure 11 entropy-21-00956-f011:**
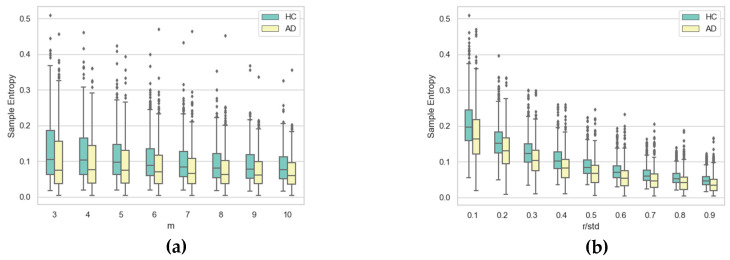
Boxplots of SE values for AD and HC considering the *y*(*t*) coordinate for (**a**) each value of *m* when varying *r* and (**b**) each value of *r* when varying *m*.

**Figure 12 entropy-21-00956-f012:**
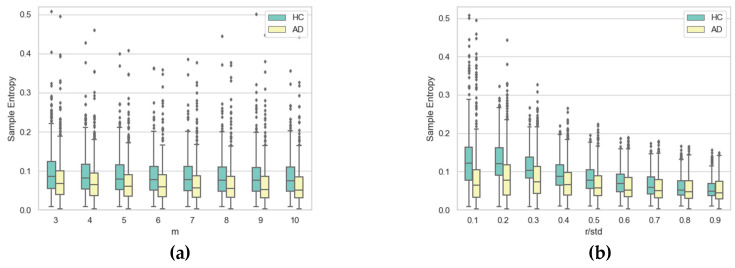
Boxplots of SE values for AD and HC considering pen pressure *p*(*t*) for (**a**) each value of *m* when varying *r* and (**b**) each value of *r* when varying *m*.

**Figure 13 entropy-21-00956-f013:**
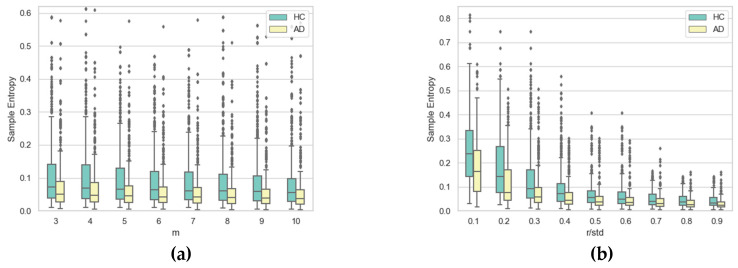
Boxplots of SE values for AD and HC considering pen altitude angle *Alt*(*t*) for (**a**) each value of *m* when varying *r* and(**b**) each value of *r* when varying *m*.

**Table 1 entropy-21-00956-t001:** Characteristics of the 70 subjects available in the dataset. HC denotes healthy controls, MMSE denotes mini-mental state examination.

	Male (n = 21)			Female (n = 49)		
	Number	Age (mean ± std)	MMSE (mean ± std)	Number	Age (mean ± std)	MMSE (mean ± std)
**AD patients** **(n = 31)**	14	81.5 ± 7.4	23.1 ± 3.8	17	78.2 ± 6.6	22.5 ± 4.3
**HC subjects** **(n = 39)**	7	75.5 ± 6.4	28.8 ± 0.7	32	72.2 ± 6.4	28.8 ± 1.1

**Table 2 entropy-21-00956-t002:** Correlation values between the MMSE and sample entropy (SE) for the five temporal functions.

MMSE, SE(*x*)	MMSE, SE(*y*)	MMSE, SE(*p*)	MMSE, SE(*Alt*)	MMSE, SE(*Az*)
0.28	0.36	0.39	0.33	0.26

**Table 3 entropy-21-00956-t003:** Correlation values between age and SE for the five temporal functions.

Age, SE(*x*)	Age, SE(*y*)	Age, SE(*p*)	Age, SE(*Alt*)	Age, SE(*Az*)
−0.22	−0.30	−0.32	−0.33	−0.10

**Table 4 entropy-21-00956-t004:** Distribution of signature styles in AD and HC groups.

Group	Stylized	Mixed	Text-Based
**AD** **(n = 31)**	21	10	0
**HC** **(n = 39)**	21	18	0

**Table 5 entropy-21-00956-t005:** Distribution of stylized and mixed signature styles in the three resulting groups.

	Group 1	Group 2	Group 3
**[mean SE(*x*), mean SE(*y*)]**	[0.2, 0.34]	[0.14, 0.23]	[0.11, 0.16]
**Stylized signatures (n = 42)**	2	30	10
**Mixed signatures (n = 28)**	0	24	4

**Table 6 entropy-21-00956-t006:** The *p*-values computed with the Mann–Whitney test for the four temporal functions.

Mann-Whitney Test	*x*(*t*)	*y*(*t*)	*p*(*t*)	*Alt*(*t*)
*p*-value	6 × 10^−5^	0.0013	3.12 × 10^−6^	4.22 × 10^−4^

**Table 7 entropy-21-00956-t007:** Classification results for the support vector machine (SVM) classifier with two-fold cross-validation.

Classification	*p*(*t*)	*Alt*(*t*)	*p*(*t*), *Alt*(*t*)	*x*(*t*),*y*(*t*),*p*(*t*),*Alt*(*t*)
Sensitivity	58.06%	61.29%	64.52%	74.19%
Specificity	76.92%	79.49%	71.79%	76.92%
Accuracy	68.80%	71.43%	68.57%	75.71%

**Table 8 entropy-21-00956-t008:** Classification results for the linear discriminant analysis (LDA) classifier with two-fold cross-validation.

Classification	*p*(*t*)	*Alt*(*t*)	*p*(*t*), *Alt*(*t*)	*x*(*t*),*y*(*t*),*p*(*t*),*Alt*(*t*)
Sensitivity	54.84%	54.84%	64.52%	67.74%
Specificity	79.49%	71.79%	79.49%	58.97%
Accuracy	68.57%	64.28%	72.86%	62.85%

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
