# Peer review of "Online Signature Analysis for Characterizing Early Stage Alzheimer’s Disease: A Feasibility Study"

_entropy, 2019, doi:10.3390/e21100956_

Round 1

Reviewer 1 Report

This work aims at exploring the online signature modality for characterizing Alzheimer’s disease (AD) at an early stage. Data from 31 AD and 39 healthy signatures were subjected to analyses of temporal sequence of pen pressure and pen altitude (vertical stroke) angle and complexity. Authors claim that  signatures made by AD patients at an early stage have lower information content (i.e. less complex) than those of healthy persons, especially in the time sequences of pen pressure and pen Altitude angle with respect to the tablet. Scores subjected to discriminant analyses for classifying AD and HC signatures had weak discriminability. 

The results of the study, as presented, do not support the authors' claims.  Moreover, the research itself has several serious flaws  - many of which can be addressed with new statistical analyses and re-interpretation of the findings; but some may require collection of more data.  An outline of the major and minor concerns follows:   

The authors propose no formal hypothesis, making it difficult to know if this study is preliminary or pilot work or whether the sample sizes were sufficient for hypothesis testing in the first place. One major limitation to the design of this study was the apparent combining of signatures of different styles (i.e. text-based, mixed, or stylized) into either the AD or HC groups. Signature styles vary significantly in complexity (and by extension sample entropy) and should be managed statistically through subgroup analyses or Multivariate ANOVAs. Given that within-group sample sizes will be low; more data may be required. It is quite possible that discriminability could improve when comparing a homogeneous signature style. A second major problem with the study design was the decision to obtain only two signatures from each subject. Two exemplars are insufficient to estimate central tendency and more importantly natural variation of an individual writer. The range of natural variation in signatures may very well overlap between AD and HC writers further weakening your conclusion that signatures made by AD patients at an early stage have lower information content than those of healthy persons. It is not clear why the authors decided to correlate age with sample entropy values and not MMSE (at least within the AD group)? If the authors hypothesize a relationship between dementia severity and lower information consent in signatures, then it makes little sense to look at age. In fact, the coefficients between age and sample entropy variables are weak at best. Are the correlations reported in table 1 from combined AD and HC subjects (n=70). This should be made clear. The conclusion reached by the authors that that AD patients exhibit a reduced ability in their fine motor skills during the signing process compared to HC due to cognitive and motor impairments induced by the neurodegenerative pathology is (lines 160-162) is not only unsupported by the data; but is misleading to the extent that early stage AD patients rarely exhibit motor impairment. AD is primarily a cognitive disorder (motor problems do emerge later in the course of the illness, however). Figures 5-8 do nothing to support any argument that AD signatures differ from HC signatures on measures of sample entropy. The conclusion most readers would reach is that sample entropy values of temporal functions (x, y, p, and Alt) do not discriminate early stage AD from healthy controls.   Authors seem to favor an alternative conclusion. Classifiers presented in Tables 2 and 3 support other data presented in this report (e.g. figures 5-8) information content is normal in AD signatures. Nonetheless, it remains possible that some AD signatures (e.g. text-based) do show reduced information content compared to HC signatures (also text-based). This analysis was not reported.

Reviewer 2 Report

This work provides interesting insights about the relationship between automatic handwriting movements and Alzheimer's disease.

The paper is clearly written, I would just suggest to double check the manuscript for English.

State of art analysis is pretty exhaustive. Authors could also mention novel classification work for PD detection through handwriting that also provides explicit rules for the clinical diagnosis (for example Senatore, Della Cioppa, Marcelli, Information 2019).

Reported results are convincing,although I would suggest to perform statistical analysis for better sustaining the claims reported in the paper and add accuracy values besides sensitivity and specificity.

Round 2

Reviewer 1 Report

Authors addressed most of the concerns from an earlier review; however, some relatively minor deficiencies remain.

Concluding that correlation coefficients on the order of 0.2 – 0.4 support a relationship between independent variables is misleading. For example, a correlation of 0.39 for the relationship between MMSE and Sample Entropy measures account for only 15% of the variance in MMSE. While this may be statistically significant (due to large sample size), it does not provide compelling support for the authors conclusion that signature complexity decreases with dementia severity. Authors should discuss these low (albeit significant) correlations in the Discussion section in a paragraph on limitations.

The report should include p-values for the coefficients in Tables 2 and 3.

The classification results reported in Table 6 suggest misclassification of approximately 25% of the AD cases. Authors should discuss this limitation and propose refinements in the study design or measures that may help to reduce this error and improve it's clinical and forensic utility.

There is no question in my mind that this is a potentially viable approach to understanding the role of cognitive decline on information contained in handwritten signatures and that the study itself was carried out with careful attention to detail and rigor. However, given the modest findings, the Discussion should be expanded to summarize limitations, the preliminary nature of the research, and future directions.
